# Genome-Wide DNA Alterations in X-Irradiated Human Gingiva Fibroblasts

**DOI:** 10.3390/ijms21165778

**Published:** 2020-08-12

**Authors:** Neetika Nath, Lisa Hagenau, Stefan Weiss, Ana Tzvetkova, Lars R. Jensen, Lars Kaderali, Matthias Port, Harry Scherthan, Andreas W. Kuss

**Affiliations:** 1Human Molecular Genetics Group, Department of Functional Genomics, Interfaculty Institute for Genetics and Functional Genomics, University Medicine Greifswald, 17475 Greifswald, Germany; neetika.nath@uni-greifswald.de (N.N.); lisa.hagenau@uni-greifswald.de (L.H.); stefan.weiss@uni-greifswald.de (S.W.); ana.tzvetkova@uni-greifswald.de (A.T.); jensenl@uni-greifswald.de (L.R.J.); 2Institute of Bioinformatics, University Medicine Greifswald, 17475 Greifswald, Germany; lars.kaderali@uni-greifswald.de; 3Bundeswehr Institute for Radiobiology Affiliated to the University of Ulm, 80937 München, Germany; MatthiasPort@bundeswehr.org (M.P.); scherth@web.de (H.S.)

**Keywords:** radiation doses, repair mechanism, translocation, transition transversion ratio, IR-induced variants, SNVs, InDels, topological associating domains

## Abstract

While ionizing radiation (IR) is a powerful tool in medical diagnostics, nuclear medicine, and radiology, it also is a serious threat to the integrity of genetic material. Mutagenic effects of IR to the human genome have long been the subject of research, yet still comparatively little is known about the genome-wide effects of IR exposure on the DNA-sequence level. In this study, we employed high throughput sequencing technologies to investigate IR-induced DNA alterations in human gingiva fibroblasts (HGF) that were acutely exposed to 0.5, 2, and 10 Gy of 240 kV X-radiation followed by repair times of 16 h or 7 days before whole-genome sequencing (WGS). Our analysis of the obtained WGS datasets revealed patterns of IR-induced variant (SNV and InDel) accumulation across the genome, within chromosomes as well as around the borders of topologically associating domains (TADs). Chromosome 19 consistently accumulated the highest SNVs and InDels events. Translocations showed variable patterns but with recurrent chromosomes of origin (e.g., Chr7 and Chr16). IR-induced InDels showed a relative increase in number relative to SNVs and a characteristic signature with respect to the frequency of triplet deletions in areas without repetitive or microhomology features. Overall experimental conditions and datasets the majority of SNVs per genome had no or little predicted functional impact with a maximum of 62, showing damaging potential. A dose-dependent effect of IR was surprisingly not apparent. We also observed a significant reduction in transition/transversion (Ti/Tv) ratios for IR-dependent SNVs, which could point to a contribution of the mismatch repair (MMR) system that strongly favors the repair of transitions over transversions, to the IR-induced DNA-damage response in human cells. Taken together, our results show the presence of distinguishable characteristic patterns of IR-induced DNA-alterations on a genome-wide level and implicate DNA-repair mechanisms in the formation of these signatures.

## 1. Introduction

Ionizing radiation (IR) plays an important role in medical diagnostics, nuclear medicine, and radiology. Radiation accidents, on the other hand, represent incidental causes of acute radiation exposures. As repeatedly seen in past events, accidental exposition to IR often involved a single acute dose of IR that led to deterministic damages such as localized injuries and acute radiation sickness [1]. Lower doses can induce stochastic effects that are the consequence of faulty repair of genomic lesions such as DNA double-strand breaks (DSBs). The resulting mutations have the potential to influence cellular functions and can alter the fate of the affected cell(s), which may also contribute to an increased risk for the affected individuals to develop cancer.

On a subcellular level, IR can either act directly on the DNA molecule by breaking one or both strands of the DNA backbone or indirectly by creating free radicals, which in turn damage the genetic material [2]. DSBs are the form of IR-induced damage with the highest biological relevance, since un- or misrepaired DSBs can lead to a loss of genomic integrity and trigger apoptosis or cancer development [3,4,5]. To prevent this, efficient and accurate cellular DNA repair mechanisms are of vital importance. DSBs are typically repaired by one of two mechanisms, non-homologous end joining (NHEJ) and homologous recombination (HR). During NHEJ, the broken ends of the DNA strands are joined back together without the use of a homologous template, whereas HR uses the homologous chromatid or chromosome as a template for error-free repair [6]. A less common repair mechanism, called microhomology-mediated end joining (MMEJ), uses microhomologous sequences to align the broken strands [7]. DNA repair can occasionally lead to chromosomal rearrangements or translocations when two fragments of different chromosomes are erroneously joined and exchanged [8]. Insertions or deletions of varying size (InDels) and single nucleotide variants (SNV) occur much more frequently than translocations. The former variants contribute substantially to genetic variation, although they can lead to a diverse range of diseases when they occur in functional regions of the DNA [9,10]. 

The advancement of DNA sequencing technology over the past years now allows the gathering of bulk information at the DNA sequence level [11] and to identify and visualize variants such as SNVs [12], InDels [13], and translocations [14] in the complete human genome sequence. In this study, we have explored a comprehensive dataset from human gingiva fibroblast (HGF) cells that were subjected to increasing X-radiation doses followed by repair intervals of 16 h and 7 days. Using the HiSeq (Illumina) technology, we sequenced the whole genome with high (60-fold) coverage in order to investigate the presence and nature of IR and repair-induced mutational signatures at the chromosomal and nucleotide level. 

## 2. Results

To investigate and gather evidence for IR-induced mutational effects in the genome sequence, we took different aspects of structural and functional features of the genome under consideration. Below we present the mutational patterns observed with respect to translocations (2.1), cytogenetic bands (2.2), and topological domains (2.4), showing specific results for chromosome 19 (2.3) as well as characteristic mutational signatures (2.5). The presented data result from two independent Next-Generation Sequencing (NGS) datasets with 60× (dataset 1) and 20× (dataset2) coverage.

### 2.1. Translocations 

First, we determined the occurrence of inter-chromosomal translocations, i.e., the exchange of chromosomal segments between different chromosomes. Figure 1 shows IR-induced translocations for two different DNA repair intervals (16 h and 7 days) in combination with increasing radiation doses (dataset 1, for dataset 2 results see Appendix A
Figure A1A).

Overall, we found 19–24 (dataset 1) and 10–16 (dataset 2) inter-chromosomal translocations after a repair interval of 16 h, and 59–90 (dataset 1) and 20–26 (dataset 2) inter-chromosomal translocations after a 7 days repair interval. This corresponds to an approximately 2- to 3-fold increase in inter-chromosomal translocation counts for all samples after a repair interval of 7 days relative to the respective repair interval of 16 h (Table 1). In general, there was no obvious dose effect. Nonetheless, for the 16 h and 7 days repair interval most translocations involve chromosome 2 in both datasets. In dataset 1, we additionally observed an accumulation of translocations involving chromosome 16 after 7 days, with the results generally being more diverse in both 7 days datasets. Overall, these results show that some chromosomes appear more sensitive to radiation than others and that prolonged time for repair and proliferation after a radiation exposure can alter the aberration spectrum seen early after the insult. 

### 2.2. Distribution of SNVs and InDels in Genome Dataset 

After sample filtering [12], i.e., the removal of sample-specific variants already present prior to IR treatment, approximately 2% of the small variants (SNVs and InDels of insertion length up to 470 nt and deletion length up to 280 nt) remained. These *de novo* IR-induced variants (mutations) were analyzed further (see also Table 1). First, we investigated the genomic accumulation and distribution of SNVs and InDels. We observed few differences in variant accumulation over dose and repair intervals (Figure 2). The maximum increase in variant accumulation was 7.4% for SNVs and 16% for InDels (Table 2) after the 16 h repair interval. There were no consistent dose response effects in our genome sequencing datasets (compare Figure 2 for dataset 1 and Appendix A
Figure A2 for dataset 2). 

When comparing the occurrence of the two types of variants (SNVs and InDels) before and after IR treatment, we observed a relative increase of InDels as compared to SNVs. The ratio of SNVs:InDels decreased from 5.17–5.37 before sample filtering to 2.04–2.27 after filtering, which represents an approximately 2.5-fold relative increase in InDels among the IR-induced variants (Table 2), a radiation specific effect.

We then looked at the numbers of IR-induced variants per chromosome. The highest variant accumulation was consistently found on chromosome 19 in both datasets and independent of dose and repair interval (Figure 2 for dataset 1 and Appendix A
Figure A2 for dataset 2). Chromosome 19 had the highest IR-specific variant accumulation, with this chromosome also showing the highest variant accumulation prior to IR also before sample filtering, i.e., the removal of sample-specific variants already present prior to IR treatment (see Appendix A
Figure A1B). However, the differences between the chromosome with the highest (Chr 19) and lowest (Chr 13) number of variants increased 5-fold regarding IR-induced variants. Overall, we saw a spread between highest and lowest variant counts by a factor of about 2 before sample filtering (i.e., in the non-irradiated genome), while this increased to a factor of 10 for the IR-induced variants (Appendix A
Figure A1B). In addition to, the ranking of the chromosomes changed with respect to their relative variant load (Compare Figure 2A and Appendix A
Figure A1B). For example, while chromosomes 4 and 6 ranked among those with the highest accumulation of variants in control samples (Appendix A
Figure A1B), subsequent sample filtering led to a much reduced relative load of IR-induced variants, placing them among the majority of chromosomes with intermediate variant accumulation (Figure 2A,B).

We then aimed to explore IR-induced SNVs and InDels in more detail and focused on the relative occurrence of transitions (Ti, A**↔**G, and C**↔**T) and transversions (Tv) on one hand, and the length distribution of the detected InDels on the other. The mean Ti/Tv ratio ranged between 1.11–2.06 (average Ti/Tv = 1.45, for dataset 1) and 1.08–1.65 (average Ti/Tv = 1.25, for dataset 2) among different samples. We failed to note any bias towards any transition or transversion event. After computing individual Ti/Tv values for each chromosome (Figure 2C for dataset 1 and Appendix A
Figure A2C for dataset 2), we observed that in dataset 1 chromosome 13 showed the highest mean Ti/Tv ratio with 2.06 at 0.5 Gy after a repair interval of 7 days. Incidentally, chromosome 13 also had the lowest variant accumulation. The lowest Ti/Tv value (dataset 1) was 1.11 on chromosome 20, which showed rather high variant accumulation.

We next analyzed the individual lengths of IR-induced InDels with respect to the reference genome sequence (hg19) by subtracting the nucleotide sequences of our dataset from the reference genome sequence (alt-ref) at the respectively corresponding positions. No significant changes in InDel length distribution were observed after X irradiation as shown in Figure 2D (for dataset 1). In general, deletions (maximum length in dataset 1: 307 nt; in dataset 2: 297 nt) were smaller than insertions (maximum length in dataset 1: 652 nt; in dataset 2: 672 nt). Mononucleotide and dinucleotide InDels were most abundant. Deletions showed a slightly less steep decrease in occurrence with increasing size than insertions (Figure 2D, dataset 1 and Appendix A
Figure A2D for dataset 2).

Next, we explored the functional impact of the IR-induced alterations by annotating these variants. Figure 2E (Appendix A
Figure A2E for dataset 2) shows functional impact information based on parameters of gene function as defined by sequence ontologies for dataset 1. Overall, the majority of SNVs were annotated as “modifiers” (which have an even smaller functional impact than the so-called “low-impact” variants) in both datasets (Table 1). High impact was predicted for 12–62 SNVs and the number of variants with moderate impact ranged from 189 to 1857 (Table 1; see also Appendix A
Table A1 for more detailed information with respect to chromosome 19). There was, however, no apparent association with dose or repair interval duration.

Previous studies have shown that gene density might play a role in chromosomal susceptibility to radiation damage and/or DNA repair priority [15,16]. Gene density, i.e., the number of genes per number of base pairs in Mb, varies considerably between chromosomes. The chromosome with the highest gene density of 22.53 genes/Mb is chromosome 19, while chromosome 13 has the lowest gene density, with 2.65 genes/Mb. To study whether the accumulation of IR-induced variants per chromosome correlates with gene density, we plotted SNV counts per Mb against gene density [17] and computed correlation scores (see Figure 2F for dataset 1, Appendix A
Figure A2F for dataset 2). These were between 0.48 and 0.59, showing a moderately positive linear relationship between gene density and SNV accumulation. This means that gene density alone does not explain the observed variance in SNV accumulation between different chromosomes. 

In summary, a clear dose responses relationship could not be obtained for the mutations in our whole genome WGS datasets. We found a relative increase in the number of InDels and observed a rise in variation of variant counts at the chromosomal level among the IR-induced variants. The observed Ti/Tv ratio was lower than the Ti/Tv ratio of the reference genome. Most of the variants had a moderate predicted functional impact and this functional impact did not follow any obvious dose effect. Variant accumulation on the other hand showed a positive correlation with gene density.

### 2.3. Intrachromosomal Distribution of SNVs and InDels

As an example for the intrachromosomal distribution and accumulation of IR-induced sequence alterations, a more detailed analysis of the mutational spectrum of the gene-rich chromosome 19 is presented, which accumulated the highest frequency of IR-induced DNA variants per Mb (see Figure 2A,B). The circular plot in Figure 3A (dataset 1) shows the distribution of SNVs and InDels across the cytogenetic bands of chromosome 19 (for dataset 2 see Appendix A
Figure A3A). A large number of variants accumulated in the pericentromeric region p12 with 303 SNVs (0.5 Gy), 295 SNVs (2 Gy), and 183 SNVs (10 Gy) 16 h after IR treatment, and 183 SNVs (0.5 Gy), 251 SNVs (2 Gy) and 240 SNVs (10 Gy) after a 7-day repair interval. For InDels in the pericentromeric region, the numbers were between 49 and 65 for all samples. In addition to that, two other cytogenetic bands (p13.3 and p13.2) on the short arm of chromosome 19 showed increased accumulation of SNVs and InDels. Interestingly, these regions differ markedly in GC-content and gene density with p13.2 being a gene-rich region with high GC content, while p12 is AT-rich and gene-poor.

Subsequently, we focused on topologically associating domains (TADs) as a functional feature of the genome [18]. TADs are self-interacting genomic regions suggesting preferential interaction between loci within these domains relative to interactions to loci outside the domain borders. TADs span from tens to thousands of kilobases and are stable across different cell types and conserved across species (for further information see e.g., [18]). These domains are not evenly distributed throughout individual chromosomes, and their borders differ from those of cytogenetic bands. We explored the distribution of variants across TADs and looked at variant distribution inside and outside the borders of TADs (Figure 3B,C for dataset 1, and Appendix A
Figure A3B,C for dataset 2). In order to do that, we binned regions inside and outside of TADs into ten equally long portions. In chromosome 19, we observed that variant accumulation was highest for both SNVs and InDels inside of the largest TAD, which corresponds to cytogenetic band p12. We also found that the variant count in chromosome 19 was always higher inside than outside of TADs with an inside:outside ratio of about 2. Furthermore, most variants were accumulated at the border of TADs.

The other chromosomes, while having accumulated smaller numbers of IR-induced alterations showed similar results with respect to variant distribution and TAD effects: On all chromosomes, band- and TAD-related patterns of variant accumulation could be observed (for more example see Appendix A
Figure A4 and Figure A5).

Next, we investigated putative functional consequences of the IR-induced variants observed on chromosome 19 as defined by sequence ontology [19], again binned over cytogenetic bands (Figure 4, dataset 1, and Appendix A
Figure A6, dataset 2). We observed only a few high impact sequence ontologies on 4 different genes on chromosome 19 in the two combined datasets (Appendix A
Table A1). Overall, the functional impact of the variants on chromosome 19 is similar to the genome-wide functional impact distribution (Table 1) and we did not find any major difference between chromosome 19 and the other chromosomes with respect to this feature.

### 2.4. InDel-specific Signatures

In order to obtain further clues as to the possible effects of the mechanisms involved in repairing IR-induced DSBs we investigated the occurrence of InDels with respect to their genomic vicinity, i.e., their occurrence in repeat regions, areas of microhomology or regions without any distinguishing features.

Figure 5 shows different short IR-induced InDel (see above) signatures as detected in dataset 1 (for dataset 2 see Appendix A
Figure A7) before and after sample filtering (See Table 3 for pertaining terms and definitions). Fold change analysis showed 1.74-fold reduction in insertion signatures after performing filtering steps. For the most part, filtering for IR-induced variants did not have an obvious effect on the distribution patterns. Interestingly, however, we observed a marked increase in the relative abundance of 3 nucleotide deletions in genomic sequence sections without repeat or microhomology, which could be considered characteristic for IR-induced InDel signatures.

## 3. Discussion

In this study, we analyzed the IR-induced mutational spectrum of IR-exposed HGFs by whole-genome sequencing. We investigated multiple aspects of IR-induced genomic alterations including translocations and distribution of small variants, and their relationship to structural and functional features of the genome.

### 3.1. Dose-Response Correlation

Interestingly, after variant calling and filtering, there was no clear dose dependency observed in either dataset both for SNVs and InDels (for dataset 1 Figure 2; Appendix A
Figure A2A,B). This contrasts with a previous exome study with a similar experimental setup where we investigated IR-induced variants in exomes of cells exposed to varying doses of X-rays and observed a dose response [12]. Such a dose response may reflect different efficacies of DNA repair pathways on different portions of the genome, of which transcription coupled base excision repair (TCR) is effectively monitoring and repairing the transcribed portion of the genome [20], while homologous recombination and NHEJ contribute to DSB repair in heterochromatin [21]. A dose-independent response on genome level, however, is not inconceivable and might reflect highly efficient DNA repair mechanisms even in cells exposed to a high dose of sparsely ionizing photon radiation. It was found that 99% of all DSBs are correctly repaired in cells after a radiation dose of 2 Gy [22] and that fetal fibroblasts undergo more efficient DNA repair after high-dose photon irradiation [23]. Since human fibroblasts are quite resistant to IR-induced apoptosis and undergo extended G1 checkpoint arrest for days after 10 Gy photon irradiation [24], our 10 Gy X irradiation will predominantly have resulted in reproductive cell death. Furthermore, cells that underwent apoptotic cell death (less than 1.5%; G. Schrock, unpublished observations; [24]) were likely lost from the sequenced population. Additionally, high dose-triggered more efficient DNA repair in fibroblasts [23] could have removed most of the damage after the 10 Gy dose. This may be one way to explain why the number of high-impact, non-lethal mutations were similar for the cells exposed to low or high dose of X radiation (Table 1). Further investigations will have to address DNA repair efficiency to elucidate DNA damage repair and load in irradiated HGFs.

### 3.2. Mutational Signatures on a Genomic Level 

In keeping with a previous study, where IR-induced DNA damage was detected as large-scale rearrangement of the genome, and sensitive chromosome regions were identified [25], we also found evidence that some chromosomes appear more sensitive to radiation than others.

For example, chromosome 2 and 16 contained the highest numbers of translocations (Figure 1). These are likely a result of DSB misrepair where broken DNA strands from different chromosomes get wrongly joined [26]. A recent study correlating chromosomal territories with IR-induced translocations in lymphocytes found that chromosomes 2 and 16 have a high number of interactions and are found in proximally positioned chromosomes [27]. It is possible that the high number of translocation events between chromosome 2 and 16 in our study also reflects spatial proximity in the nucleus. In addition, chromosome 16 has already been shown to be radiation sensitive, which might explain the high number of translocations for its small size [28]. This also correlates with high SNV and InDel rates observed in our study (Figure 2A,B).

We found a lower ratio of SNVs:InDels i.e., a relative increase in InDels, in both datasets when comparing variant counts before and after removing non-IR-induced variants (see Table 2). In a previous study, Adewoye et al. analyzed germline mutation rates induced in mice after parental exposure to IR and found that the rate of induction of *de novo* InDels compared to controls was significantly increased, while the rate of SNVs was not significantly elevated [13]. InDels accumulation after IR treatments is likely caused by additional DSBs induced by IR and has been observed in mice, IR-associated tumor cells, and induced pluripotent stem cells (iPSCs) [13,29,30,31]. Taken together, this suggests a radiation-specific effect.

When looking at the SNV-type spectrum after filtering at the chromosomal level we did not see any bias towards a specific transition or transversion event (Figure 2C). However, we did observe an IR-induced reduction in Ti/Tv ratios after filtering. In the human genome, the mean Ti/Tv ratio is approximately 2.1, while we calculated a mean Ti/Tv ratio of 1.45 for IR-induced SNVs across all samples. This is in line with different studies in plants and mammals [32,33], who report a conspicuously high proportion of transversions after irradiation. Moreover, it is consistent with our observations in exome data from HGF cells that received an identical treatment as the cells used in this study [34]. A high frequency of transversions are considered a feature of oxidative DNA damage [35] and could here indicate the indirect effect of radiation through hydroxyl radicals by radiolysis of water [6]. On the other hand, mismatch repair (MMR) also leads to an increased number of transversions [36]. While MMR is usually coupled to DNA replication and highly effective in repairing replication errors, several studies have shown cell cycle-independent recruitment of MMR proteins to DNA lesions after treatment with UV light and alkylating agents and propose a mutagenic role for MMR in certain contexts [37,38].

### 3.3. Intrachromosomal IR-Related Mutation Patterns

We showed that the accumulation of both SNVs and InDels after irradiation differs considerably between chromosomes (Figure 2A,B) and while there are differences in the numbers of mutations per chromosome in the non-irradiated controls, the variance there is not as high as in the irradiated samples. This means that some chromosomes are more affected by ionizing radiation than others.

Chromosome 19 showed the highest load of IR-induced alterations under all experimental circumstances, while Chromosome 13 showed substantially less mutational damage. InDels, for example, occur as much as 10 times more often on chromosome 19 as compared to chromosome 13. As these chromosomes are on different ends of the gene density spectrum, we investigated whether the mutational accumulation is correlated to gene density in the whole genome. We observed a moderate positive correlation for gene density and mutation accumulation in our data (Figure 2F), which is in line with previous research, showing that gene-rich chromosomes are more susceptible to DNA damage [16,39]. 

But also on an intrachromosomal level, patterns of mutation accumulation can be observed in the individual chromosomes. For example, we observed the highest overall rate of SNVs and InDels in Chr19 p12, adjacent to the centromere (Figure 3), where also 1 SNV with putatively deleterious effect is located (affecting ZNF93). This region is relatively gene-poor with a low GC content (40.7%) and a high density of repeat elements and pseudogenes [40] as well as a large cluster of genes encoding zinc finger transcription factors, which are exclusive to primates [41,42]. This contrasts with the finding that gene density is a moderately good predictor for mutation accumulation. Apart from gene density, transcriptional activity and chromatin structure have also been suggested to influence DNA damage and repair [43,44]. It is thus likely that only a combination of various factors will be able to fully explain the differential mutation accumulation patterns/signatures in our data.

Previously, it has been discussed that IR-induced lesions can also have an impact on the 3D organization of the genome and recent results from Hi-C experiments show a decrease in interactions across TAD boundaries after exposure to IR in human fibroblasts [14]. The authors suggest that this could play a role in the ability of cells to maintain genome integrity and prevent deleterious translocations across TAD boundaries [14]. In this study, we evaluated the distribution of IR-induced alterations inside and outside of TADs and found that both SNVs and InDels inside of TADS tend to accumulate towards the TAD boundaries in most chromosomes (see also Appendix A
Figure A4 and Figure A5). In mammals, TAD boundaries show enrichment of CCCTC-binding factor (CTCF) binding (for recent review see e.g., [45]), which increases with higher insulation scores [14]. Increased CTCF recruitment to TAD boundaries as a consequence of IR treatment could, therefore, be expected to result in enhanced DNA repair in these areas, as CTCF was found to determine the nanostructure of the γ-H2AX DSB repair domains [46] and to facilitate DNA double-strand break repair by enhancing homologous recombination repair [47,48]. However, it is not yet fully understood in which way CTCF exerts its various functions. It may thus be speculated that CTCF at the bases of TADs will provide a genome-wide distributed pool of chromatin-bound CTCF. If a DSB occurs, CTCF will keep the chromatin loops in its vicinity from flaring out and even facilitate or participate in their repair by its role in HR repair of DSBs. 

So, it may be speculated that CTCF binding to the flanking chromatin loops around a DSB will lead to close accumulation of DSB-flanking DNA, thereby facilitating its repair, which is further enhanced given that CTCF is functioning in HR repair. Thus, IR-dependent recruitment of CTCF to TADs may be considered to contribute to the IR-dependent mutation pattern observed.

Interestingly, we consistently observed a relative increase in triplet deletions among IR-induced InDels in areas without repetitive or microhomologous features. This points to a so far unexplored bias of the involved DNA-repair mechanisms in these contexts and might indicate the possibility of an increased tolerance towards in-frame deletions. Taken together, our results reveal distinguishable and characteristic patterns of IR-induced DNA-alterations on a genome-wide level. In keeping with previous findings from studies in *C. elegans* [47] we come to the conclusion that these signatures are the joint product of DNA damage, repair and tolerance.

## 4. Materials and Methods 

### 4.1. Cell Culture and IR Treatment

Primary HGFs were obtained from Provitro AG (Berlin, Germany), cultured, and treated as previously described [49]. Confluent cell cultures were irradiated with 240 kV X rays at 13 mA (YXLON Maxishot, Hamburg, Germany) filtered with 3 mm beryllium at a dose rate of 1 Gy min^−1^. Exposure to 0, 0.5, 2 and 10 Gy was combined with two repair intervals (16 h and 7 days, see also Table 2).

### 4.2. Next-Generation Sequencing Analysis

DNA was prepared using the NucleoSpin^®^ Tissue Kit from Macherey-Nagel (Düren, Germany) and used for NGS library preparation and sequencing. Two independent datasets with 60× (dataset 1) and 20× coverage (dataset 2) were generated, which were sequenced by the BGI Tech Solutions Co. (Hong Kong), and Max Planck Institute for plant breeding research (Cologne, Germany) sequencing facilities, respectively. One Sample (10Gy/7d) from dataset 2 had a 10 × coverage.

### 4.3. Bioinformatics Analysis

The bioinformatics processing was performed as follows: Paired-end reads obtained through WGS were trimmed using trimmomatric followed by mapping using tophat and CUSHAW to the hg19 (GRCh37) version of the human reference genome. After mapping, we adapted variant calling steps from Genome Analysis Toolkit (GATK) for version 3.3. The corresponding variants were assessed using R version 3.6.1 to perform filtering and further analysis as previously published (for further details see [12,31].

### 4.4. Data Filtering 

In our analysis, we have implemented three filtering steps: (1) sample filtering, (2) coverage filtering and (3) Binomial filtering. For sample filtering, we used variant pools from untreated control samples for both repair intervals to remove IR-independent and cell line-specific variants [41]. For coverage filtering, we used our previously published parameter settings, of a sequencing depth (DP) value of 3 and genotype quality (GQ) value of 20 [31]. For binomial filtering, we have considered an error rate of 0.001 to remove the artifact that is due to sequencing (see again also [31]).

### 4.5. Translocation Analysis

Translocation analysis was conducted using BreakDancer version 1.4.5 [50] to predict interchromosomal translocations from paired-end sequencing reads for each sample. As suggested by the manual, we used BreakDancer’s bam2cfg.pl to create the parameter configuration for each sample following the structural variant analysis using breakdancer-max to determine translocations. For each sample, we removed translocations that were also present in the control samples to keep only translocations that were induced by the repair mechanism after radiation of cells. Translocations were defined as equal if they were present within the same 200-nucleotide window. We then only kept translocations containing at least 20 supported reads as well as a confidence score greater than 80 percent.

### 4.6. Variant Accumulation Analysis

Variant (V) accumulation analysis was performed for chromosomal bands and TADs Variant counts per Mb were normalized over chromosomal band length using:V/Mb = 1,000,000 × (Count of V per cytogenetic band)/(Total length of cytogenetic band)(1)

For TAD-based analyses we used the human TAD coordinates from the Hi-C project [18]. We downloaded the ‘TADs in the hg19′ folder from the 3D Genome Browser (http://promoter.bx.psu.edu/hi-c/publications.html, accessed on 4 June 2020) that actualizes the TAD coordinates from the Dixon et al. [18] pipeline. To ensure a close match with the cell type used in our study we choose the file “IMR90_Lieberman-raw_TADs”, which was generated based on fibroblasts.

### 4.7. SNV-based analyses

SNVs were subdivided into transitions and transversions to investigate the different substitution types. Transitions refer to a pyrimidine-pyrimidine (C↔T) or purine-purine substitution (A↔G), whereas transversions refer to pyrimidine-purine substitutions or vice versa (A↔C, A↔T, C↔G, G↔T). To compute the transition (Ti) and transversion (Tv) ratio:nTi/nTv = (Transition count/Transversion count)(2)

### 4.8. InDel-Based Analyses

InDel counts were further subdivided into numbers of insertions and deletions; nucleotides were counted in the reference genome sequence as “ref” and in the corresponding position in the genome sequence generated in this study as “alt”. If count (ref) > count(alt) a deletion was called, while count(ref) < count(alt) was considered an insertion.

In order to identify microhomologous mutational signatures for InDels in our dataset, we use the adapted functions in the R package called “mutSigExtractor” [7]. For InDel signatures, the definitions are dependent on signatures/contexts counts within the data. Basically, InDel signatures were classified first as insertions or deletions, second, according to the length of possible flanking mononucleotide repeats, and third, possible areas of microhomology at InDel boundaries. See Table 3 for the terms used for InDel classification and average log fold changes for the respective alterations.

### 4.9. Functional Annotation 

VCF files were annotated with SnpEff and then processed to summarize the contained sequence ontology information. By default, SnpEff uses the Ensembl transcripts to search for genomic features overlapping the location of the variants. Thus, it is often possible that a specific variant overlaps the same gene ontological feature from different transcripts. In such cases, we counted the feature type for the chromosomal location only once. As the Ensembl transcripts are quite numerous it is also possible to have several different feature types for the same location. To reduce background noise by counting of annotation more than once we used only the protein-coding transcripts included in the “known” Reference Sequence (RefSeq) collection. Finally, we combined some of the features into less specific ones (Table 4).

## Figures and Tables

**Figure 1 ijms-21-05778-f001:**
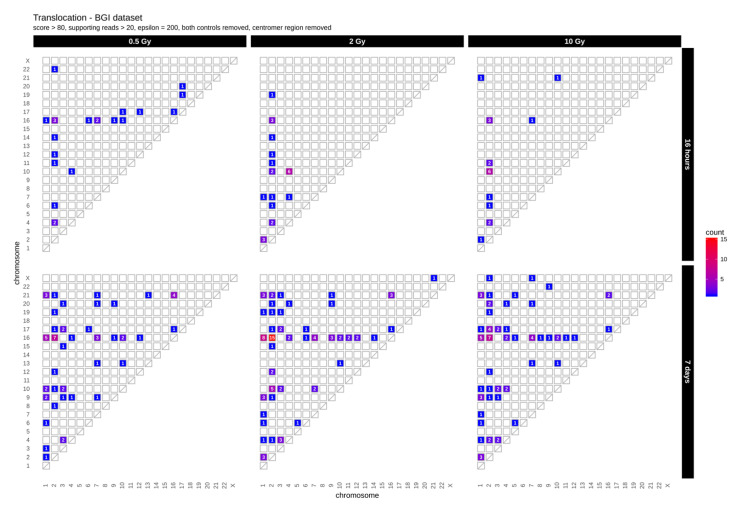
Translocation counts between chromosomes (irrespective of the direction) in genome dataset 1 (generated with BreakDancer).

**Figure 2 ijms-21-05778-f002:**
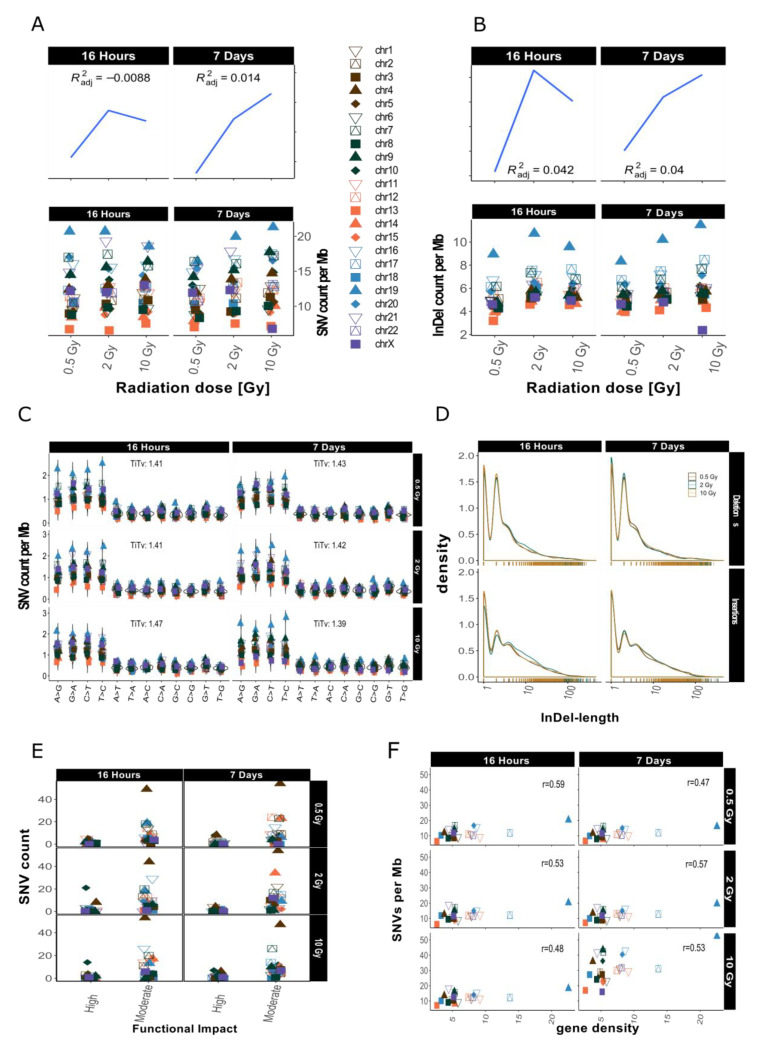
Overview for IR-induced SNV accumulation in genome dataset 1 (for genome dataset 2 see Appendix A
Figure A2). Accumulation of SNVs per Mb ((**A**), lower panel) and InDels per Mb ((**B**), lower panel) per chromosome is shown together with the corresponding trends with adjusted root mean square (upper panels). (**C**): SNV types, i.e., as transitions (Ti) and transversions (Tv) per MB per chromosome and Ti/Tv values for each sample. (**D**): Distribution of insertions and deletions with respect to size. (**E**): SNV count per chromosome with respect to functional impact (HIGH and MODERATE functional consequences only). (**F**): SNVs per Mb per chromosome with respect to relative chromosomal gene density. Gene density was computed based on hg19 (GRCh37) using the custom gene set (excluding predicted genes) described b [17].

**Figure 3 ijms-21-05778-f003:**
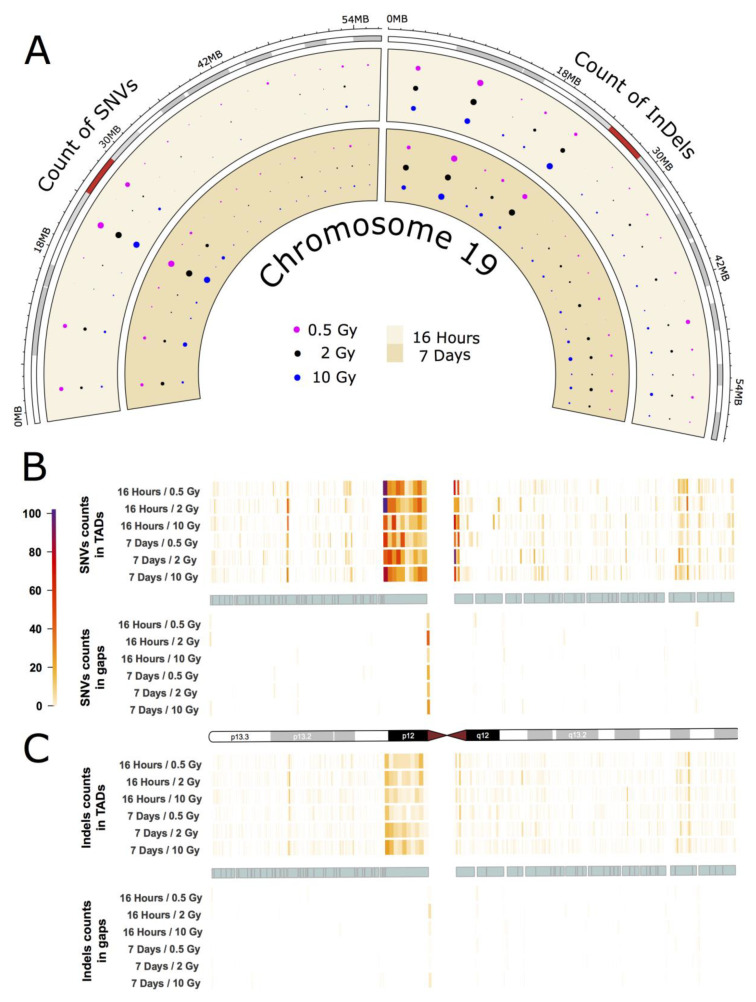
IR-induced variant accumulation across chromosome 19 (dataset 1). (**A**): Distribution of SNVs and InDels over cytogenetic bands. Topologically associating domain (TAD) plot showing the distribution of SNV (**B**) and InDel (**C**) counts inside and outside of TAD coordinates for chromosome 19. The chromosome is represented in the middle of the figure as an ideogram. Each panel is subdivided, showing the SNV and InDel alteration counts inside (above) and outside of TADs (below). The TADs are shown as cyan blocks symmetrically in both panels. Each TAD and each gap (region outside TAD termed as gap) was binned into 10 equally long portions and the counts of the SNVs or InDels were then calculated for each portion. The heatmaps with color gradients from white through orange and red to dark purple represent values ranging from 0 to 104 of variant counts.

**Figure 4 ijms-21-05778-f004:**
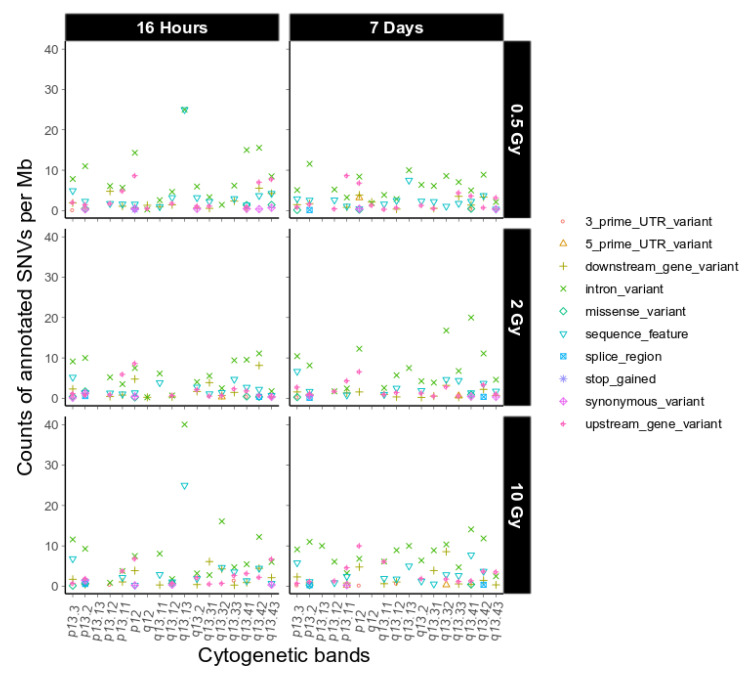
Sequence ontological information for the IR-induced variants observed on chromosome 19 over cytogenetic band, filtered for refseq IDs for dataset 1. For further explanation of the used combination of sequence ontology terms see Section 4.9. Functional Annotation.

**Figure 5 ijms-21-05778-f005:**
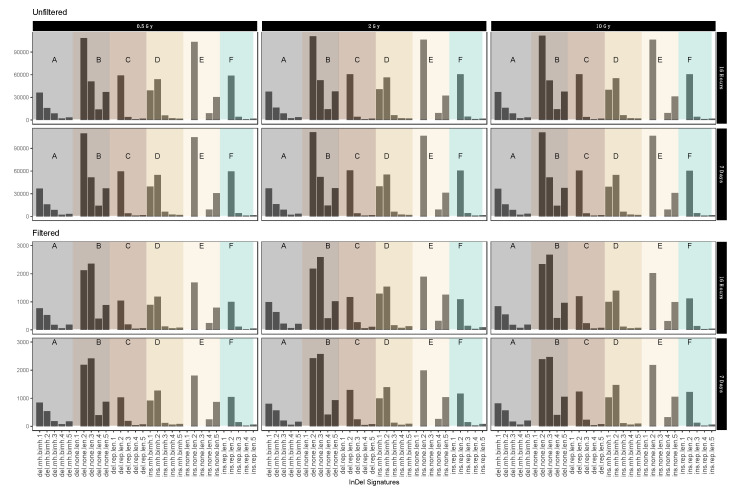
Mutational IR-induced signature plot (dataset 1). On the X-axis, the length and specific vicinity of InDels are represented, while the Y-axis (note differing scales) shows the InDel count. See Table 3 for pertaining terms and definitions.

**Table 1 ijms-21-05778-t001:** Counts of IR-induced variants (sequence annotation with SNPEff annotator) as well as translocations for all samples.

Time	Radiation Dose	Functional Impacts				Inter-Chromosomal Translocations
Dataset 1		High	Moderate	Low	Modifier	
	0.5 Gy	14	189	3785	33,384	22
16 h	2 Gy	21	226	3800	35,714	24
	10 Gy	15	222	3984	35,514	19
	0.5 Gy	18	216	3408	33,189	59
7 Days	2 Gy	12	263	3717	35,336	90
	10 Gy	22	208	3980	36,754	74
Dataset 2						
	0.5 Gy	19	277	4849	50,437	14
16 h	2 Gy	49	262	5119	52,567	10
	10 Gy	30	432	6107	60,561	16
	0.5 Gy	62	1857	14,227	76,636	23
7 Days	2 Gy	36	475	7416	69,550	26
	10 Gy	40	210	11,140	68,032	20

**Table 2 ijms-21-05778-t002:** Table reporting the depth (DP) and variant counts after and (before) filtering steps.

Time	Radiation Dose	SNVs Depth (DP) After (Before) Filter	InDels Depth (DP) After (Before) Filter	SNVs Counts After (Before) Filter	InDels Counts After (Before) Filter
Dataset 1					
16 h	0.5 Gy	1,029,599 (206,573,706)	334,619 (25,449,747)	34,261 (3,839,076)	15,083 (715,406)
	2 Gy	1,098,772 (221,379,800)	372,440 (27,846,456)	36,674 (3,859,535)	17,982 (739,759)
	10 Gy	1,101,857 (222,284,695)	379,065 (28,064,422)	36,435 (3,854,999)	17,098 (734,831)
7 Days	0.5 Gy	1,030,578 (217,184,675)	352,673 (26,948,643)	34,078 (3,846,812)	15,698 (724,272)
	2 Gy	1,092,867 (217,841,944)	369,254 (27,518,060)	36,256 (3,850,694)	17,188 (734,633)
	10 Gy	1,142,613 (236,100,648)	397,153 (29,973,246)	36,836 (3,778,441)	17,518 (730,883)
Dataset 2					
16 h	0.5 Gy	1,166,694 (130,621,022)	242,653 (6,073,404)	50,516 (3,848,840)	16,987 (252,782)
	2 Gy	1,137,296 (123,863,517)	234,141 (5,764,483)	52,666 (3,852,339)	17,675 (254,521)
	10 Gy	1,474,964 (153,573,569)	351,819 (7,637,043)	60,688 (3,901,398)	25,620 (269,267)
7 Days	0.5 Gy	2,268,357 (187,652,991)	478,893 (9,473,714)	77,818 (3,953,911)	29,045 (274,854)
	2 Gy	1,825,231 (17,5864,812)	432,240 (8,902,319)	69,691 (3,930,167)	27,463 (272,142)
	10 Gy	387,902 (84,432,203)	266,508 (1,973,176)	27,251 (3,510,867)	28,387 (91,240)

**Table 3 ijms-21-05778-t003:** Terms and definitions used for mutational signatures in Figure 5 and Appendix A
Figure A7 with average fold change from unfiltered and filtered data.

Terms	Definition	Average Log Fold Change Form Filtered to Unfiltered
Del.mh.bimh.x (A)	Deletion in area of microhomology	−1.66
Del.none.len.x (B)	Deletion in undistinguished area	−1.69
Del.rep.len.x (C)	Deletion in repeat area	−1.72
Ins.mh.bimh.x (D)	Insertion in area of microhomology	−1.70
Ins.none.len.x (E)	Insertion in undistinguished area	−1.74
Ins.rep.len.x (F)	Insertion in repeat area	−1.74

**Table 4 ijms-21-05778-t004:** Definitions of combined sequence ontology terms used in this study.

SnpEff Sequence Ontology	Combined Terms
splice_region_variant&intron_variant; splice_region_variant; splice_region_variant&synonymous_variant; splice_region_variant&non_coding_exon_variant	splice_region
splice_acceptor_variant&intron_variant; splice_donor_variant&intron_variant; splice_acceptor_variant&splice_donor_variant&intron_variant	splice_site
missense_variant&splice_region_variant	missense_variant
stop_retained_variant	synonymous_variant
stop_gained&splice_region_variant	stop_gained
5_prime_UTR_premature_start_codon_gain_variant	5_prime_UTR_variant

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
