# Peer review of "Genome-Wide DNA Alterations in X-Irradiated Human Gingiva Fibroblasts"

_ijms, 2020, doi:10.3390/ijms21165778_

Round 1

Reviewer 1 Report

Professor Kuss and the coauthors introduced their discovery in the genomic alternation in human Gingiva fibroblasts cause by ionic radiation. They evaluated the levels of translocation between chromosomes, mutations, inserts and deletions which response to the ionic radiation. The results are interesting and I would like to recommend the manuscript to be published once all the minor questions have been addressed.

  1. The difference of changing fold between data set 1 and set 2 is noticeable. The counts summarized in set 2 does not seem support the conclusion that both chromosome 2 and 16 presented significantly more translocation. A third data set helps greatly to make a more solid conclusion.
  2. In Page 4 section 2.2, please use a consistent description for the IR treatment since readers may be confused it with data filtering.
  3. Page 5, Line 124. Please specifically identify the samples, does it refer to samples from different does, repair interval, data sets, or chromosomes?
  4. Page 8, figure3A. Please use an image in higher resolution since it is hard to read the numbers for the locations.
  5. Page11, section 3.1. It is very interesting to have a does-dependent effect in exome while there is no does-dependent effect over the whole genome. If the accurate repair efficiently rescue the damage caused by radiation, does this repair work discriminate the DNA region between transcribed area and nontranscribed area? However, it doesn't seem reasonable that the repair is less efficient in exome since damages accumulated in transcribed regions and these regions are more sensitive to irradiation. I am just curious and authors' interpretation is reasonable as well.

Author Response

Professor Kuss and the coauthors introduced their discovery in the genomic alternation in human Gingiva fibroblasts cause by ionic radiation. They evaluated the levels of translocation between chromosomes, mutations, inserts and deletions which response to the ionic radiation. The results are interesting and I would like to recommend the manuscript to be published once all the minor questions have been addressed.

The difference of changing fold between data set 1 and set 2 is noticeable. The counts summarized in set 2 does not seem support the conclusion that both chromosome 2 and 16 presented significantly more translocation. A third data set helps greatly to make a more solid conclusion.

>> We thank the reviewer for pointing this out. We noticed the highest numbers of translocations involving chromosome 2 in both datasets but 16 only in dataset 1. We have corrected this inaccuracy in the representation of our results. The pertaining statement now reads “Nonetheless, for the 16-hour and 7 day repair interval most translocations involve chromosome 2 in both datasets. In dataset 1, we additionally observed an accumulation of translocations involving chromosome 16 after 7 days, with the results generally being more diverse in both 7d datasets.” (p.3 l.90-93)

In Page 4 section 2.2, please use a consistent description for the IR treatment since readers may be confused it with data filtering.

>> We are grateful for this comment and have changed this section (p.4 l.99-169) to clarify the distinction between the IR-induced variants and variants that were present prior to IR – treatment (and subsequently removed through our filtering process).

Page 5, Line 124. Please specifically identify the samples, does it refer to samples from different does, repair interval, data sets, or chromosomes?

>> We have altered the text accordingly (p.5 l.131-133).

Page 8, figure3A. Please use an image in higher resolution since it is hard to read the numbers for the locations.

>> We have now made the labelling legible.

Page11, section 3.1. It is very interesting to have a does-dependent effect in exome while there is no does-dependent effect over the whole genome. If the accurate repair efficiently rescue the damage caused by radiation, does this repair work discriminate the DNA region between transcribed area and nontranscribed area? However, it doesn't seem reasonable that the repair is less efficient in exome since damages accumulated in transcribed regions and these regions are more sensitive to irradiation. I am just curious and authors' interpretation is reasonable as well.

>> We appreciate the reviewers interest and agree that this question is an important cue for further investigations, requiring more exome datasets. We have also expanded this section of the discussion:

“Interestingly, after variant calling and filtering, there was no clear dose dependency observed in either dataset both for SNVs and InDels (for dataset 1 Figure 2; Fig. A2, A and B). This contrasts with a previous exome study with a similar experimental setup where we investigated IR-induced variants in exomes of cells exposed to varying doses of X-rays and observed a dose response [12]. Such a dose response may reflect different efficacies of DNA repair pathways on different portions of the genome, of which transcription coupled base excision repair (TCR) is effectively monitoring and repairing the transcribed portion of the genome [20] while homologous recombination and NHEJ contribute to DSB repair in heterochromatin [21]. A dose-independent response on genome level, however, is not inconceivable and might reflect highly efficient DNA repair mechanisms even in cells exposed to a high dose of sparsely ionizing photon radiation. It was found that 99% of all DSBs are correctly repaired in healthy HGF cells after a radiation dose of 2 Gy [22] and that fetal fibroblasts undergo more efficient DNA repair after high dose photon irradiation [23]. Since human fibroblasts are quite resistant to IR-induced apoptosis and undergo extended G1 checkpoint arrest for days after 10 Gy photon irradiation [24] our X IR will predominantly have resulted in reproductive cell death. Furthermore, cells that underwent apoptotic cell death (less than 1.5%; G.Schrock, unpublished observations; [24]) were likely lost from the sequenced population. Additionally, high dose-triggered more efficient DNA repair [23]  could have removed most of the damage after the 10Gy dose. This may explain why the number of high-impact, non-lethal mutations were similar for the cells exposed to low or high dose of X radiation (impact table Table 1). Further investigations will have to address DNA repair efficiency to elucidate DNA damage repair and load in irradiated HGFs. (p.11-12 l.255-275).

Reviewer 2 Report

The manuscript presents an interesting set of experimental data on the genome-wide in vitro effects of exposure to ionizing radiation. In my opinion, there are a number of issues which have not properly been properly described in the text.

  1. Given that the authors have sequenced DNA samples extracted from the highly heterogeneous ‘population’ of irradiated cells, the key question is – what was the sensitivity of the technique used in this study? From the text it does not follow what was the frequency of mutations found in this study. This important issue should be clarified. Besides, the authors should present the distribution of mutation frequencies for all classes of mutants detected in their study.
  2. The good question is – what about the frequency of spontaneous somatic mutation in these cells? Without such an information it remains unclear what was the frequency of radiation-induced mutation in the exposed cells. It would appear to me that a substantial proportion of de novo mutations described in this study may be attribute to spontaneous mutational events.
  3. It also does not follow from the text whether the authors have attempted to validate some of mutation candidates detected by the next-generation sequencing. If so, then another question is – what was the frequency of false positives in this study?

Minor issues:

  1. I would advise the authors to add in the Results section the definition of datasets (20x and 60x coverage).
  2. In the legend to Fig. 1 should the authors mention what dataset was used (I presume the 60x coverage).
  3. In Table 1 the authors should add nan extra row to the top explaining that High/Moderate/Low columns show the functional impact of mutants.

Author Response

The manuscript presents an interesting set of experimental data on the genome-wide in vitro effects of exposure to ionizing radiation. In my opinion, there are a number of issues which have not properly been properly described in the text.

Given that the authors have sequenced DNA samples extracted from the highly heterogeneous ‘population’ of irradiated cells, the key question is – what was the sensitivity of the technique used in this study? From the text it does not follow what was the frequency of mutations found in this study. This important issue should be clarified. Besides, the authors should present the distribution of mutation frequencies for all classes of mutants detected in their study.

>> Thank you! In order to clarify, which technique was used we now make an according to statement in the introduction, so that the reader is aware of what to expect with respect to sensitivity and accuracy: “The presented data result from two independent Next Generation Sequencing (NGS) datasets with 60x (dataset 1) and 20x (dataset 2) coverage.” (p.2 l.76-77).

Concerning the frequency of alterations, we would like to refer to table 2 where we present absolute counts before and after filtering. As these results are presented per sample they could be considered to represent the frequencies of variants per genome. We have, however, refrained from using this term because it is problematic as we analyzed cell populations and not individual genomes. Single cell sequencing would be much better suited to answer this question but is beyond the scope of this study.

The good question is – what about the frequency of spontaneous somatic mutation in these cells? Without such an information it remains unclear what was the frequency of radiation-induced mutation in the exposed cells. It would appear to me that a substantial proportion of de novo mutations described in this study may be attribute to spontaneous mutational events.

>> In a recent publication, the mutation rate of skin fibroblasts was determined to be 1.57x10^-7 mutations/basepair/cell division (Werner, B., Sottoriva, A., 2018. Variation of mutational burden in healthy human tissues suggests non-random strand segregation and allows measuring somatic mutation rates. PLoS Comput. Biol. 14. https://doi.org/10.1371/journal.pcbi.1006233). With the human genome containing roughly 3x10^9 basepairs, this would result in approx 500 mutations. We identified between about 40,000 and 50,000 IR-induced mutations (SNV and InDels) in the data, i.e. after removal of variants that the cell line contained before irradiation (see Table1). The number of spontaneous mutations would thus account for approx. 1% of the detected IR-induced alterations. As the cells were grown to confluency before irradiation, it is unlikely that cells were replicating in the 16h repair interval samples. The number of expected spontaneous mutations could be slightly higher in the 7day repair interval samples as some cells would possibly start to replicate but still not to an extent that we would consider meaningful.

It also does not follow from the text whether the authors have attempted to validate some of mutation candidates detected by the next-generation sequencing. If so, then another question is – what was the frequency of false positives in this study?

>> As radiation induced mutations occur randomly throughout the genomes of the irradiated cells, the focus of our work was to identify IR-induced genome wide alteration patterns rather than individual mutations. Therefore, we did not validate individual mutations and used a very stringent and previously evaluated set of filters (see Nath et al. 2018 Exome Sequencing Discloses Ionizing-radiation-induced DNA Variants in the Genome of Human Gingiva Fibroblasts Health Phys. 115(1):151-160 https://pubmed.ncbi.nlm.nih.gov/29787441/) to minimize the number of possible false positives.

Minor issues:

I would advise the authors to add in the Results section the definition of datasets (20x and 60x coverage).

>> See answer to first comment of Reviewer 2.

In the legend to Fig. 1 should the authors mention what dataset was used (I presume the 60x coverage).

In Table 1 the authors should add an extra row to the top explaining that High/Moderate/Low columns show the functional impact of mutants.

>> We have made the according changes pertaining to these two issues and also revised the figures/legends again for better legibility and clarity.

Round 2

Reviewer 2 Report

In the revised manuscript the authors have addressed all my comments. I am happy to recommend this MS for publication.